# Assessment of Psychological and Social Fitness in Healthy Adults Permanently Living at Very High Altitude

**DOI:** 10.3390/ijerph20032013

**Published:** 2023-01-21

**Authors:** Chun Gao, Jizong Ciren, Dan Wang, Zhaohui Zhang, Ruidong Ge, Li’e Yan

**Affiliations:** 1Department of Gastroenterology, China-Japan Friendship Hospital, Beijing 100029, China; 2Research Center for Physical Fitness at High Altitude, Public Cultural Service Center, Chengguan District Culture and Tourism Bureau, Lhasa 850000, China; 3Evaluation Research Center, Renmin University of China, Beijing 100872, China; 4Department of Rehabilitation Medicine, China-Japan Friendship Hospital, Beijing 100029, China; 5Nursing Department, China-Japan Friendship Hospital, Beijing 100029, China

**Keywords:** psychological fitness, social fitness, high altitude, healthy adults

## Abstract

Background: Environmental factors of high altitude, especially hypobaric hypoxia, may directly and persistently affect human physical and mental health. Our study was designed to assess the psychological and social fitness in healthy adults permanently living at very high altitude, i.e., an average elevation of 3650 m. Methods: In our observational study, 320 participants were included, among which 218 (68.1%) had resided in such a setting for more than 20 years. Participants underwent 138 assessments, including the Self-Rating Anxiety Scale (SAS), Symptom Check List 90 (SCL-90) and the Evaluation Scale of Human Adaptation Capability (ESHAC). SAS (20 items) and SCL-90 (90 items) were used to assess psychological fitness, and the ESHAC (28 items) was used to assess social fitness. Pearson analysis was used to assess correlations and Logistic regression analysis was performed to determine independent influencing factors. Results: The highest SAS score was 80 and the mean score was 43.26 ± 8.88, which was higher than the norm in China (*p* < 0.001). Sixty (18.8%) participants showed anxiety symptoms and 14 (4.4%) had moderate or severe anxiety. The average score of SCL-90 was 140.88 ± 44.77, and 96 (30.0%) participants showed SCL-90 scores ≥160. Compared with the norm, significant differences were shown in eight of the nine SCL-90 factor scores, i.e., somatization, obsessive-compulsive, depression, anxiety, hostility, phobic anxiety, paranoid ideation, and psychoticism. The average score of ESHAC was 19.92 ± 4.54, and 114 (35.6%) participants did not reach the qualifying standard. Significant correlations were observed between the SAS score, SCL-90 total and factor scores, and ESHAC scores. The Logistic regression analysis showed that being born at very high altitude was an independent influencing factor (AOR = 2.619; 95% CI, 1.629–4.211; *p* < 0.001) after controlling for other factors. Conclusion: Permanently living at very high altitude can influence the psychological and social fitness of healthy adults.

## 1. Introduction

High altitudes (HA), defined as 2500 m or more above sea level, are characterized by lower atmospheric oxygen pressure, increased ultraviolet radiation exposure, cold climate and extremely low absolute humidity [1,2]. These factors may directly and persistently affect human physical and mental health; hypobaric hypoxia is the main driving factor of a series of physiological and pathological changes [2,3]. If the latitude remains unchanged, each elevation of 100 m in altitude can lead to a decrease in atmospheric pressure of 5.9 mmHg and a decrease in the partial pressure of oxygen of 1.2 mmHg [4]. At altitudes of 2500–3000 m, humans usually begin to present the symptoms of hypoxia, including increased heart rate, accelerated breathing, headache, sleep disturbance, loss of appetite, decreased exercise ability, and even altitude illness [4]. Life-threatening complications include severe mountain sickness, high-altitude pulmonary edema and high-altitude cerebral edema [5].

Health has been redefined by the World Health Organization as a state of complete physical, mental and social well-being, and not merely the absence of disease or infirmity [6]. This definition presupposes that physical, mental and social well-being are the three main dimensions of health [7,8], and that mental health is an integral and essential component of health. Mental health is more than the absence of mental disorders; it is also determined by socioeconomic, environmental and biological factors [8]. Recently, the notion of good health has been extended to include five dimensions, namely: physical fitness, emotional fitness, social fitness, spiritual fitness and cultural fitness [7]. Living at a high altitude and hypobaric hypoxia may lead to physical inactivity, which has been associated with cognitive decline, anxiety, depression, increased prevalence of chronic diseases, and a significant decline in functional capacity [9].

Psychological and social fitness are two important components of health. They can be assessed using a number of evaluation scales, for example, Symptom Check List 90 (SCL-90), Self-Rating Anxiety Scale (SAS), Self-Rating Depression Scale (SDS), Social Adaptation Self-evaluation Scale (SASS) and Vineland Adaptive Behavior Scales (VABS) [10,11,12]. Considering that anxiety was the most commonly observed disorder [13,14], the SAS and SCL-90 were used to assess psychological fitness, and the Evaluation Scale of Human Adaptation Capability (ESHAC) was used to assess social fitness in our study. The aims of our research were to assess the psychological and social fitness of healthy adults permanently living at very high altitude and to assess the influence of high altitude on these parameters.

## 2. Materials and Methods

### 2.1. Study Population

Thirteen cultural stations run by neighborhood committees and the Public Cultural Service Center affiliated with the Chengguan District Culture and Tourism Bureau provide public cultural services for the residents living in all the communities in Lhasa City, the capital of the Tibet Autonomous Region. Chengguan District has 473,586 inhabitants. During the period between July 2021 and June 2022, 341 people were contacted to partake in our observational study. A total of 320 participants who participated in cultural activities actively and followed the inclusion criteria were included in the final analysis. The inclusion criteria were as follows: (1) older than 20 years and less than 59 years of age; (2) absence of severe diseases of major organs, including chronic heart failure, chronic respiratory failure, end stage liver disease, uremia and advanced malignant tumors, as determined by interviews conducted by the researchers; (3) no affective disorders, according to their medical history and medication history; (4) permanent residence in Lhasa City for at least three years, with an average elevation of 3650 m; and (5) sufficient educational level to correctly understand the meanings of three assessment scales. The participants filled in the questionnaires on paper independently and only one time. Completion of the questionnaire took less than one hour. Our research strictly followed the Declaration of Helsinki, and the written informed consent was obtained from all the participants.

### 2.2. Zung Self-Rating Anxiety Scale (SAS) Assessment

Both the SAS, which was developed by Zung in 1971, and the Symptom Check List 90 (SCL-90) assessments were used to assess the psychological fitness of our participants in this study, considering that anxiety was the most commonly observed disorder [13,14]. Twenty items are included in the SAS, among which 15 are scored in a positive manner and five in a negative manner. For each item, a scale from 1 to 4 is used, based on the degree of a given symptom. The SAS score (the total score is 100 points) is computed as follows: the scores of the 20 items are added together, the sum is multiplied by 1.25, and decimals are rounded to the nearest whole number. The evaluative standard is as follows: ≥50 is indicative of anxiety symptoms, whereby 50–59 indicates mild anxiety, 60–69 indicates moderate anxiety, and ≥70 points indicates severe anxiety.

### 2.3. Symptom Check List 90 (SCL-90) Assessment

The SCL-90 questionnaire, which was proposed by Derogatis, was used to further assess psychological fitness based on the SAS assessment. A total of 90 items are included in the SCL-90 assessment. A scale of 1–5 is used for each item, and the maximum score is 450 points. Evaluative standard: An SCL-90 score ≥160 is indicates symptoms of psychopathology and psychological distress. The 10 factors of SCL-90 are somatization (12 items), obsessive-compulsive (10 items), interpersonal sensitivity (9 items), depression (13 items), anxiety (10 items), hostility (6 items), phobic anxiety (7 items), paranoid ideation (6 items), psychoticism (10 items) and other (7 items). For each SCL-90 factor, the average score is calculated and the total factor score is 5 points. The results of SCL-90 in our study were compared with the norm in China.

### 2.4. The Evaluation Scale of Human Adaptation Capability (ESHAC)

The social fitness of our participants was assessed using the ESHAC, which was developed by Hong Ren and issued at the 9th National Sports Science Conference in China in 2011. Twenty-eight items are included in the ESHAC, which contains three subscales: adaptation to the natural environment (9 items), adaptation to the social environment (10 items) and adaptation to physiological change and resistance to diseases (9 items). Each item is rated on a ten-point scale (from 1 to 10); the maximal scores of the three subscales are 90, 100 and 90 points, respectively. To simplify the analysis, the final scores of the three subscales are the total scores divided by the number of items, and the final total score is 10 points for each subscale. The ESHAC score is the sum of the scores of the three subscales; therefore, the highest possible score is 30 points. An ESHAC score ≥18 is considered to be the qualifying standard.

### 2.5. Statistical Analysis

SPSS for Windows, version 21.0 (SPSS, Chicago, IL, USA) was used for statistical analyses. The mean ± standard deviation and Independent-Samples T-test were used for continuous variables. The numbers (proportions) and Pearson Chi-Square test were used for categorical variables. The Single-Sample T-test was used to compare the SAS and SCL-90 scores of our participants with Chinese norms. Subgroup analyses were performed and the 320 participants were subgrouped according to gender, age, ethnicity, birthplace, marital status and duration of residence in Lhasa City. Pearson analysis was used to assess the correlation between the SAS standard score, SCL-90 total score, SCL-90 factor scores, ESHAC total score and the scores in the three subscales of the ESHAC. Logistic regression analysis was performed to determine the independent influencing factors for psychological and social fitness. The dependent variable was performance in the psychological and social fitness assessments, and six independent variables were analyzed. Participants who met one of the following criteria were considered to have poor psychological and social fitness: (1) SAS score ≥ 50; (2) SCL-90 score ≥ 160; and (3) ESHAC score < 18. The results are expressed as odds ratios (ORs) or adjusted odds ratios (AORs) and their 95% confidence intervals (CIs).

## 3. Results

### 3.1. Demographic Characteristics

Three hundred and twenty healthy participants were included in our study; their demographic characteristics are shown in Table 1. Ages ranged from 20 to 59 years old, and the mean age was 30.5 ± 8.2 years. Of the study group, 60.0% (192 people) was male and 40.0% female. Tibetan (Zang) is the main nationality in the Tibet Autonomous Region of China, accounting for 60.6% of our study population. Among all 320 participants, 64.7% were born in the Tibet Autonomous Region while 35.3% moved to this snow-covered plateau. For the duration of residence in Lhasa City, 218 (68.1%) participants had lived there for more than twenty years.

### 3.2. Psychological Fitness: The Incidence and Severity of Anxiety as Assessed by SAS

Anxiety was the most common disorder among mental illnesses [13,14]. The SAS questionnaire was used to assess the incidence and severity of anxiety for our evaluation of psychological fitness. Of our 320 participants, the highest SAS score was 80; the mean score was 43.26 ± 8.88 points. The norm for SAS in China was measured by the Scale Collaboration Group [15], which included 1158 Chinese people; the average score was 29.78 ± 0.46 points. Compared with the norm in China, statistical significance was found (*p* < 0.001), which indicated that the SAS score of our population was higher than the norm. For the incidence and severity of anxiety, 60 (18.8%) participants were assessed to have anxiety symptoms, including 46 (14.4%) with mild anxiety, 10 (3.1%) with moderate anxiety and 4 (1.3%) with severe anxiety. A comparative analysis of 60 participants with anxiety and 260 participants without anxiety is shown in Table 2; no statistical significances were found.

### 3.3. Psychological Fitness: The Results of SCL-90 Assessment

Based on the SAS assessment, SCL-90 was used to further assess psychological fitness. The results are shown in Table 3. The highest SCL-90 score among our participants was 377, and the average score was 140.88 ± 44.77 (the highest possible score is 450 points). According to the evaluative standard for the occurrence of positive symptoms, 96 participants showed SCL-90 score ≥ 160, representing 30.0% of the sample. The norm of SCL-90 in China has been measured [16,17] and the SCL-90 factor scores were compared between our participants and the national norm group. There were significant differences in eight of the nine SCL-90 factor scores, i.e., somatization (1.50 ± 0.50 vs. 1.37 ± 0.48, *p* < 0.001), obsessive-compulsive (1.80 ± 0.62 vs. 1.62 ± 0.58, *p* < 0.001), depression (1.59 ± 0.59 vs. 1.50 ± 0.59, *p* = 0.004), anxiety (1.50 ± 0.54 vs. 1.39 ± 0.43, *p* < 0.001), hostility (1.58 ± 0.62 vs. 1.46 ± 0.55, *p* < 0.001), phobic anxiety (1.36 ± 0.51 vs. 1.23 ± 0.41, *p* < 0.001), paranoid ideation (1.52 ± 0.59 vs. 1.43 ± 0.57, *p* = 0.004) and psychoticism (1.48 ± 0.52 vs. 1.29 ± 0.42, *p* < 0.001). Compared with the norm in China, higher scores were observed in our participants.

A comparison of the total score of SCL-90 and the factor scores between males and females, between participants aged 20–29 and 30–59 years, and between Tibetan and non-Tibetan are shown in Table 3. No statistical significances were found in the total SCL-90 scores in these comparisons. For the factor scores, compared with female participants, males had lower scores in somatization (1.45 ± 0.49 vs. 1.59 ± 0.49, *p* = 0.012), obsessive-compulsive (1.70 ± 0.60 vs. 1.95 ± 0.61, *p* < 0.001) and depression (1.53 ± 0.56 vs. 1.70 ± 0.61, *p* = 0.012). Participants aged 20–29 years showed higher scores in phobic anxiety (1.43 ± 0.57 vs. 1.29 ± 0.42, *p* = 0.018) than those aged 30–59 years, and the Tibetan group had higher scores in phobic anxiety (1.42 ± 0.54 vs. 1.28 ± 0.46, *p* = 0.020) than those with non-Tibetan nationalities. No significant differences were found in the other comparisons.

### 3.4. Social Fitness: Results of Evaluation Scale of Human Adaptation Capability (ESHAC)

The ESHAC assessment may be used to assess the social fitness. It contains three subscales: adaptation to the natural environment, adaptation to the social environment and adaptation to the physiological environment (Table 4). Among our participants, the lowest ESHAC score was 5.69, and the average score was 19.92 ± 4.54 points (the total score is 30). One hundred and fourteen (35.6%) participants did not reach the qualifying standard (ESHAC score ≥18); this percentage was higher than that in the SAS and SCL-90 assessments. The highest possible score in each subscale is 10 points; the average scores in the three subscales were 6.76 ± 1.77, 6.79 ± 1.68 and 6.37 ± 1.82, respectively.

Subgroup analyses were performed, and the results are shown in Table 4. The 320 participants were sub-grouped according to the gender, age, ethnicity, birthplace, marital status and duration of residence in Lhasa City. There were significant differences in ethnicity (Tibetan, 19.34 ± 4.80 vs. non-Tibetan, 20.81 ± 3.95, *p* = 0.003), birthplace (Born in the Tibet Autonomous Region, 19.16 ± 4.71 vs. Not born in the Tibet, 21.31 ± 3.85, *p* < 0.001) and duration of residence in Lhasa (Time >30 years, 18.93 ± 4.44 vs. Time ≤30 years, 20.35 ± 4.52, *p* = 0.010). For the three subscales, compared with Tibetans, the non-Tibetans had higher scores in the adaptation to natural environment (*p* < 0.001) and adaptation to physiological environment (*p* = 0.006) categories; however, no statistical significance was observed in the adaptation to social environment (*p* = 0.248) category. Similar results were shown for birthplace and duration of residence.

### 3.5. Correlation between Psychological and Social Fitness: Pearson Analysis

Pearson analysis was used to assess the correlation between psychological and social fitness. As shown in Table 5, a positive correlation between the SAS standard score and SCL-90 total score was observed (r = 0.528, *p* < 0.001), whereas negative correlations were found between the SAS score and the ESHAC total score (r = −0.172, *p* = 0.002), and between the SCL-90 total score and the ESHAC score (r = −0.380, *p* < 0.001). Moreover, significant correlations were shown between the SAS score, nine SCL-90 factor scores and the scores in three subscales of the ESHAC.

### 3.6. Factors Influencing Psychological and Social Fitness: Logistic Regression Analysis

Logistic regression analysis was performed to determine the independent influencing factors of psychological and social fitness. Based on the results of SAS, SCL-90 and ESHAC, the 320 participants were divided into two groups, comprising 162 (50.6%) participants who had at least one positive result among the three assessments and the remaining 158 (49.4%) participants who had no positive result and served as the control group. The dependent variable was the psychological and social fitness assessments, and six independent variables were analyzed, namely, gender, age, ethnicity, birthplace, marital status and duration of residence (Table 6). Univariate analysis revealed statistical significances in the Tibetan nationality (OR = 1.967; 95% CI, 1.247–3.101; *p* = 0.004) and born at very high altitude (OR = 2.619; 95% CI, 1.629–4.211; *p* < 0.001) cohorts. Multivariable analysis showed that born at very high altitude was an independent influencing factor (AOR = 2.619; 95% CI, 1.629–4.211; *p* < 0.001) after controlling for other factors.

## 4. Discussion

Lhasa City is situated at the Qinghai–Tibet Plateau, which as the highest plateau on Earth, has long been known as the roof of the world. Permanently living at very high altitude may affect human physical and mental health [2]. Our study was designed to assess the psychological and social fitness levels of healthy adults residing in Lhasa City, with an average elevation of 3650 m. The results revealed that for psychological fitness, our 320 participants had scores in the SAS and eight out of the nine SCL-90 factors that were above the Chinese norms. For social fitness, 114 (35.6%) participants did not reach the qualifying standard. Taken together, 162 (50.6%) participants had at least one positive result among the three assessments and were assessed to have poor psychological and social fitness. Moreover, our Logistic regression analysis showed that being born at very high altitude was an independent influencing factor (AOR = 2.619; 95% CI, 1.629–4.211).

Mental health is one of the important dimensions of overall health. Indeed, as stated by the World Health Organization, “there is no health without mental health” [8]. Mental health is associated with the ability to work productively, to cope with the normal stresses of life, to make good use of one’s own abilities, and to make a contribution to one’s community [8]. The first strength of our study was that we focused on mental and social well-being, and that psychological and social fitness were treated as a whole. The second was that healthy adults were selected as the research object; people with severe diseases of major organs were excluded. The third was that our 320 participants had lived in Lhasa City (Qinghai-Tibet Plateau) for at least three years, and 218 (68.1%) of them had resided there for more than twenty years.

SAS and SCL-90 questionnaires were used to assess the psychological fitness of our participants. Considering that anxiety is the most common disorder among mental diseases [13,14], the SAS can not only be used to assess the incidence and severity of anxiety but also be used to validate the results of SCL-90 assessments. As shown in Table 5, significant correlations were observed between the SAS score and the SCL-90 total and factor scores, which was completely in line with the expectations. The SAS score of our participants was 43.26 ± 8.88, which was higher than the norm in China. We compared the result with that in other populations [18,19]. In one study, which was designed to explore the alterations of functional connectivity in subregions of the basal forebrain, 96 healthy controls were included and the SAS score was 33.46 ± 2.69 [18]. Another study aimed to investigate the prevalence of anxiety among medical students in Inner Mongolia (China); the average SAS score of 1187 students was 39.60 ± 7.81 [19].

The SCL-90 assessment in our study showed that the total score was 140.88 ± 44.77; the scores in eight of the nine factors were higher than the norms. Moreover, statistical significances were found in some SCL-90 factor scores between males and females, between participants aged 20–29 and 30–59 years and between Tibetans and non-Tibetans. Our results were compared with those obtained from other researchers [20,21]. One study included 200 healthy volunteers; the results showed that the scores in somatization, obsessive-compulsive, interpersonal sensitivity, depression, anxiety, hostility, phobia, paranoid ideation and psychoticism were 1.13 ± 0.13, 1.29 ± 0.27, 1.31 ± 0.21, 1.26 ± 0.33, 1.21 ± 0.21, 1.08 ± 0.26, 1.05 ± 0.18, 1.12 ± 0.23 and 1.17 ± 0.26, respectively [20]. All of the factor scores for our participants were higher than those in that study. Another study found differences in depression and hostility between the males and females and differences in obsessive-compulsive, anxiety and phobia among different age groups [21]. We also found that the Tibetan group had higher scores in phobia than the non-Tibetan group. Further study is required to provide an explanation for this finding.

The social fitness of our participants was assessed by the ESHAC, which was developed by a Chinese scholar. We believed that this questionnaire is particularly suited to Chinese populations. Firstly, the positive results in the SCL-90 and ESHAC were compared. Ninety-six (30.0%) participants showed SCL-90 score ≥ 160, and 114 (35.6%) did not reach the qualifying standard. The two rates were very close, and no significant difference was found. Secondly, the correlation between the ESHAC, SAS and SCL-90 scores was determined by Pearson analysis. Significant correlations were shown between the ESHAC total score, scores in three subscales of ESHAC, the SAS score and nine SCL-90 factor scores. In addition, subgroup analyses were performed and significant differences in ethnicity, birthplace and duration of residence were observed. Further analysis showed that the differences were mainly due to the adaptation to the natural and physiological environment, as opposed to the social environment.

Influencing factors of poor psychological and social fitness were determined by a Logistic regression analysis. The results showed that being born at very high altitude is an independent influencing factor (AOR = 2.619; 95% CI, 1.629–4.211), after controlling for other factors. One study from China included 644 older adults and found that cognitive impairment was more common in people living at high altitude [22]. Another cross-sectional study conducted in Colombia was designed to assess the association between quality of life and municipality altitude. Six domains were measured: physical health, psychological health, social relations, economic status, functional status and medical history [23]. The results showed that high altitude was associated with lower quality of life, i.e., social relations (OR = 2.16 95%CI 1.73–2.70), physical health (OR = 1.92, 95%CI 1.47–2.52), psychological health (OR = 1.59, 95%CI 1.26–2.00) and functional status (OR = 1.80, 95%CI 1.45–2.23) [23]. Taken together, these results show that psychological and social fitness might be impaired in people living at high altitude.

For marital status, 130 (40.6%) participants lived without a spouse or partner; this proportion seemed to be higher than that of the general population. The main reason for this may have been that more young people who actively participated in cultural activities were included in our study; the mean age of our cohort was 30.5 ± 8.2 years. This study was intended to promote the safety of participants living at very high altitude, although our approach may have led to a selection bias. For the differences between male and female participants, no statistical significances were found in the SAS score, SCL-90 total score or ESHAC scores. However, for the SCL-90 factor scores, males had lower scores in somatization, obsessive-compulsive and depression than female participants.

## 5. Conclusions

Our study showed that permanently living at very high altitude may influence the psychological and social fitness of otherwise healthy adults. Psychological and social fitness are the two important components of health, and they also need the same attention as physical fitness. In future, some effective strategies should be implemented to improve the psychological and social fitness in people residing at high altitude.

## Figures and Tables

**Table 1 ijerph-20-02013-t001:** Demographic characteristics of the study population.

Characteristics	Study Population, n = 320
Gender	
Male sex, no. (%)	192 (60.0)
Female sex, no. (%)	128 (40.0)
Age	
Mean age ^#^, years	30.5 ± 8.2
20–29 years, no. (%)	169 (52.8)
30–39 years, no. (%)	97 (30.3)
40–59 years, no. (%)	54 (16.9)
Ethnicity	
Tibetan nationality, no. (%)	194 (60.6)
non-Tibetan nationality, no. (%)	126 (39.4)
Born in the Tibet Autonomous Region	
Yes, no. (%)	207 (64.7)
No, no. (%)	113 (35.3)
Marital status	
Living with a spouse or partner, no. (%)	190 (59.4)
Without a spouse or partner, no. (%)	130 (40.6)
Duration of residence in Lhasa City (Tibet)	
>30 years, no. (%)	98 (30.6)
20–30 years, no. (%)	120 (37.5)
<20 years, no. (%)	102 (31.9)

^#^ Plus-minus value indicates mean ± standard deviation.

**Table 2 ijerph-20-02013-t002:** Psychological fitness: the incidence of anxiety and a comparative analysis of 320 participants with and without anxiety.

Characteristics	Total Participants, n = 320	Participants with Anxiety (n = 60)	Participants without Anxiety (n = 260)	*p* Value
SAS standard score ^#^	43.26 ± 8.88	56.85 ± 7.33	40.13 ± 5.68	
Positive rate (SAS score ≥50) ^§^, no. (%)	60 (18.8)	60 (100.0)	0 (0.0)	
Gender				
Male sex, no. (%)	192 (60.0)	31 (51.7)	161 (61.9)	0.144
Female sex, no. (%)	128 (40.0)	29 (48.3)	99 (38.1)	
Age				
20–29 years, no. (%)	169 (52.8)	35 (58.3)	134 (51.5)	0.342
30–59 years, no. (%)	151 (47.2)	25 (41.7)	126 (48.5)	
Ethnicity				
Tibetan nationality, no. (%)	194 (60.6)	37 (61.7)	157 (60.4)	0.855
non-Tibetan nationality, no. (%)	126 (39.4)	23 (38.3)	103 (39.6)	
Born in Tibet Autonomous Region				
Yes, no. (%)	207 (64.7)	41 (68.3)	166 (63.8)	0.512
No, no. (%)	113 (35.3)	19 (31.7)	94 (36.2)	
Marital status				
With a spouse or partner, no. (%)	190 (59.4)	29 (48.3)	161 (61.9)	0.053
Without a spouse or partner, no. (%)	130 (40.6)	31 (51.7)	99 (38.1)	
Duration of residence in Lhasa City				
>30 years, no. (%)	98 (30.6)	17 (28.3)	81 (31.2)	0.669
≤30 years, no. (%)	222 (69.4)	43 (71.7)	179 (68.8)	

^#^ Plus-minus value indicates mean ± standard deviation. ^§^ The total standard score of Zung Self-Rating Anxiety Scale (SAS) is 100 and the SAS score ≥50 is assessed to be positive.

**Table 3 ijerph-20-02013-t003:** Psychological fitness: results of the SCL-90 assessment.

SCL-90 Assessment	Participants, n = 320	Gender		Age		Ethnicity	
		Male	Female	20–29 Years	30–59 Years	Tibetan	Non-Tibetan
SCL-90 total score ^#^	140.88 ± 44.77	137.02 ± 45.36	146.67 ± 43.40	141.36 ± 46.00	140.34 ± 43.50	142.65 ± 46.10	138.14 ± 42.67
Positive rate (≥160) ^§^, no. (%)	96 (30.0)	53 (27.6)	43 (33.6)	54 (32.0)	42 (27.8)	65 (33.5)	31 (24.6)
SCL-90 factor score ^#^							
Somatization	1.50 ± 0.50	1.45 ± 0.49	1.59 ± 0.49 *	1.49 ± 0.52	1.52 ± 0.47	1.54 ± 0.52	1.45 ± 0.45
Obsessive-compulsive	1.80 ± 0.62	1.70 ± 0.60	1.95 ± 0.61 **	1.80 ± 0.61	1.81 ± 0.62	1.79 ± 0.61	1.83 ± 0.62
Interpersonal sensitivity	1.60 ± 0.59	1.56 ± 0.60	1.65 ± 0.57	1.62 ± 0.60	1.57 ± 0.58	1.63 ± 0.59	1.56 ± 0.59
Depression	1.59 ± 0.59	1.53 ± 0.56	1.70 ± 0.61 *	1.59 ± 0.61	1.60 ± 0.57	1.59 ± 0.57	1.60 ± 0.62
Anxiety	1.50 ± 0.54	1.46 ± 0.54	1.57 ± 0.53	1.51 ± 0.54	1.50 ± 0.54	1.51 ± 0.56	1.49 ± 0.50
Hostility	1.58 ± 0.62	1.58 ± 0.65	1.59 ± 0.57	1.59 ± 0.63	1.58 ± 0.61	1.63 ± 0.64	1.52 ± 0.59
Phobic anxiety	1.36 ± 0.51	1.32 ± 0.51	1.42 ± 0.51	1.43 ± 0.57	1.29 ± 0.42 *	1.42 ± 0.54	1.28 ± 0.46 *
Paranoid ideation	1.52 ± 0.59	1.52 ± 0.61	1.53 ± 0.56	1.53 ± 0.60	1.52 ± 0.58	1.57 ± 0.62	1.46 ± 0.53
Psychoticism	1.48 ± 0.52	1.47 ± 0.52	1.49 ± 0.52	1.48 ± 0.52	1.47 ± 0.52	1.51 ± 0.55	1.42 ± 0.47
Other	1.67 ± 0.59	1.64 ± 0.58	1.71 ± 0.60	1.64 ± 0.58	1.70 ± 0.59	1.67 ± 0.62	1.66 ± 0.52

* *p* < 0.05. ** *p* < 0.01. ^#^ Plus-minus value indicates mean ± standard deviation. ^§^ The total score of Symptom Check List 90 (SCL-90) is 450 and the SCL-90 score ≥160 is assessed to be positive.

**Table 4 ijerph-20-02013-t004:** Social fitness: the results of the Evaluation Scale of Human Adaptation Capability (ESHAC).

Characteristics ^#^	ESHAC Total Score ^§^	*p* Value	Three Subscales		
			To Natural Environment	To Social Environment	To Physiological Environment
Gender					
Male sex	20.20 ± 4.77	0.180	6.81 ± 1.79	6.88 ± 1.76	6.51 ± 1.80
Female sex	19.50 ± 4.16		6.68 ± 1.74	6.66 ± 1.55	6.16 ± 1.84
Age					
20–29 years	20.11 ± 4.74	0.425	6.83 ± 1.81	6.75 ± 1.76	6.53 ± 1.75
30–59 years	19.71 ± 4.30		6.67 ± 1.72	6.84 ± 1.58	6.19 ± 1.88
Ethnicity					
Tibetan nationality	19.34 ± 4.80	0.003	6.49 ± 1.91 **	6.71 ± 1.74	6.15 ± 1.87 **
non-Tibetan nationality	20.81 ± 3.95		7.17 ± 1.43	6.93 ± 1.57	6.71 ± 1.68
Born in Tibet Autonomous Region					
Yes	19.16 ± 4.71	<0.001	6.43 ± 1.87 **	6.63 ± 1.73 *	6.11 ± 1.86 **
No	21.31 ± 3.85		7.36 ± 1.39	7.09 ± 1.53	6.86 ± 1.65
Marital status					
With a spouse or partner	19.94 ± 4.61	0.916	6.73 ± 1.81	6.84 ± 1.72	6.37 ± 1.80
Without a spouse or partner	19.89 ± 4.46		6.80 ± 1.71	6.72 ± 1.62	6.37 ± 1.85
Duration of residence in Lhasa City					
>30 years	18.93 ± 4.44	0.010	6.34 ± 1.80 **	6.70 ± 1.66	5.90 ± 1.92 **
≤30 years	20.35 ± 4.52		6.94 ± 1.73	6.83 ± 1.69	6.58 ± 1.74

* *p* < 0.05. ** *p* < 0.01. ^#^ Plus-minus value indicates mean ± standard deviation. ^§^ The Evaluation Scale of Human Adaptation Capability (ESHAC) contains three subscales: adaptation to the natural environment, adaptation to the social environment and adaptation to physiological change and resistance to diseases. The highest possible score in each subscale is 10; the highest possible score in the ESHAC assessment is 30; an ESHAC score <18 is considered to be positive.

**Table 5 ijerph-20-02013-t005:** Correlation between psychological and social fitness: Pearson analysis.

Correlation Analysis	SAS Standard Score	ESHAC Total Score	Three Subscales of ESHAC		
			To Natural Environment	To Social Environment	To Physiological Environment
SAS standard score		−0.172 **	−0.124 *	−0.127 *	−0.190 **
SCL-90 total score	0.528 **	−0.380 **	−0.271 **	−0.401 **	−0.315 **
SCL-90 factor score					
Somatization	0.528 **	−0.334 **	−0.263 **	−0.284 **	−0.317 **
Obsessive-compulsive	0.441 **	−0.302 **	−0.194 **	−0.329 **	−0.262 **
Interpersonal sensitivity	0.434 **	−0.334 **	−0.214 **	−0.405 **	−0.252 **
Depression	0.540 **	−0.384 **	−0.262 **	−0.418 **	−0.319 **
Anxiety	0.527 **	−0.316 **	−0.222 **	−0.345 **	−0.256 **
Hostility	0.456 **	−0.300 **	−0.175 **	−0.321 **	−0.284 **
Phobic anxiety	0.456 **	−0.335 **	−0.301 **	−0.341 **	−0.230 **
Paranoid ideation	0.368 **	−0.305 **	−0.202 **	−0.352 **	−0.240 **
Psychoticism	0.443 **	−0.359 **	−0.285 **	−0.390 **	−0.260 **
Other	0.385 **	−0.366 **	−0.277 **	−0.335 **	−0.336 **

ESHAC = Evaluation Scale of Human Adaptation Capability. SAS = Self-Rating Anxiety Scale. SCL-90 = Symptom Check List 90. * *p* < 0.05. ** *p* < 0.01.

**Table 6 ijerph-20-02013-t006:** Influencing factors for psychological and social fitness: Logistic regression analysis.

Variable	Univariate Analysis			Multivariable Analysis ^#^		
	OR	95% CI	*p* Value	AOR	95% CI	*p* Value
Gender: Male sex	0.803	0.513–1.257	0.338			
Age: 30–59 years	0.930	0.600–1.443	0.746			
Ethnicity: Tibetan nationality	1.967	1.247–3.101	0.004	0.830	0.382–1.800	0.637
Born in Tibet Autonomous Region	2.619	1.629–4.211	<0.001	2.619	1.629–4.211	<0.001
Living with a spouse or partner	0.764	0.488–1.195	0.238			
Duration of residence >30 years	1.459	0.904–2.356	0.122			

AOR = adjusted odds ratio. OR = odds ratio. CI = confidence interval. ^#^ Logistic regression analysis was performed to determine the independent influencing factors. Participants who met one of these criteria were considered to demonstrate poor psychological and social fitness: (1) SAS score ≥50; (2) SCL-90 score ≥160; and (3) ESHAC score <18.

## Data Availability

Data available on request from the authors.

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
