# Peer review of "Assessment of Psychological and Social Fitness in Healthy Adults Permanently Living at Very High Altitude"

_ijerph, 2023, doi:10.3390/ijerph20032013_

Round 1

Reviewer 1 Report

I appreciate the work that the authors did. The present study is about people living at very high altitudes and how this is affecting mental health. The findings from such studies are relevant for the population, the health care services in place, and occupational medicine since some professions are practiced at these altitudes.

This is an observational study. I suggest the atuhors to use the checklist of STROBE to improve the manuscript.

You can find some of my suggestions in the PDF.

I have some major remarks about the methods that are not mentioned in the PDF file:

In which and over what period of time did the investigation take place? Was it during summer or winter time (which year)?

I would like to get more information about the recruitment procedure.

Do you have a flow chart about the recruitment procedure?

Do you have a figure with inclusion and exclusion criteria?

How many people were contacted to partake in the study?

How many people live in the study area?

How many people were excluded from the study?

How was the health status of the study participants assessed?

What do you mean by “(2) they thought they had been in good condition of health, and they had no severe diseases of major organs, including heart, lung, liver and kidney”?

This wording is too vague for me. Could it please be explained a little more precisely, e.g., the participants were asked about possible pre-existing conditions by questionnaire? Or the participants were examined for possible pre-existing conditions or interviewed by the study conductor?

Please define severe diseases.

Did you include persons with affective disorders?

How did the participants fill in the questionnaires? Was it online or on paper? Did you provide them with help? How many times did they answer the questionnaires, only one time or several times?

I think this is an interesting study. But the description of methods (study design, setting) should be improved.

Reviewer 2 Report

The idea of this article is novel. The influence of plateau factors on people's mental and psychological factors and adaptability were analyzed. The statistical method was used properly and the conclusion was reliable.

There are several questions: 1. How is the patient sample size calculated?2. Is there any difference between the sexes in this research conclusion? Suggestion: Accept it after minor repairs.

Reviewer 3 Report

The authors carried an interesting research on influence of the psychological and social fitness in healthy adults permanently living at very high altitude, and showed that  born at very high altitude was an independent influence factor.   The article is well-structured and has a unique point of view. There are two questions that need to be answered.

1.Participants with anxiety were diagnosed with the SAS standard score. So the first p value is not necessary in table 1. 

2.As to marital status, the participants without a spouse or partner have a high proportion. When participants were enrolled, was it consistent with the overall proportion, or did  a selection bias exist?

Reviewer 4 Report

It is a very well designed study, congratulations!

Round 2

Reviewer 1 Report

It is very nice that you considered my recommendations and that you used the STROBE Guideline. 

I wish you all the best with your publication.